# Needle-Tract Seeding of Pancreatic Cancer after EUS-FNA: A Systematic Review of Case Reports and Discussion of Management

**DOI:** 10.3390/cancers14246130

**Published:** 2022-12-12

**Authors:** Livia Archibugi, Ruggero Ponz de Leon Pisani, Maria Chiara Petrone, Gianpaolo Balzano, Massimo Falconi, Claudio Doglioni, Gabriele Capurso, Paolo Giorgio Arcidiacono

**Affiliations:** 1Pancreato-Biliary Endoscopy and Endosonography Division, Pancreas Translational & Clinical Research Center, IRCCS San Raffaele Scientific Institute, Vita-Salute San Raffaele University, Via Olgettina 60, 20132 Milan, Italy; 2Pancreatic Surgery Unit, Pancreas Translational & Clinical Research Center, IRCCS San Raffaele Scientific Institute, Vita-Salute San Raffaele University, Via Olgettina 60, 20132 Milan, Italy; 3Pathology Department, IRCCS San Raffaele Scientific Institute, Vita-Salute San Raffaele University, Via Olgettina 60, 20132 Milan, Italy

**Keywords:** pancreatic cancer, needle tract seeding, endoscopic ultrasound, fine needle aspiration, pancreatic adenocarcinoma

## Abstract

**Simple Summary:**

Needle-tract seeding (NTS) is a rare but serious complication of Endoscopic Ultrasound-guided biopsy of pancreatic adenocarcinoma. Due to its very low incidence, there is lack of evidence about its management. The aim of our systematic review is to deepen the knowledge about this entity and highlight therapeutic approaches. After a systematic search we retrieved 45 cases, plus one from our center. The majority of patients (87.1%) underwent surgical resection of the primary pancreatic tumor, of which only 55.9% received neoadjuvant and/or adjuvant chemotherapy. NTS occurred at a median of 19 months after primary diagnosis, with a secondary surgical approach in 82.4% of patients and a median overall survival of 26.5 months compared to 15.5 months when treated with chemo/radiotherapy. NTS is rare, generally occurs late and might be treated aggressively with good results. As only a low rate of patients developing NTS underwent (neo)adjuvant chemotherapy, one may speculate that it might be protective.

**Abstract:**

Needle-tract seeding (NTS) has been sporadically reported as complication of Endoscopic Ultrasound (EUS)-guided aspiration (FNA) in pancreatic adenocarcinoma (PDAC). However, the evidence of its treatment and outcome is sparse. Adhering to PRISMA guidelines, we conducted a systematic review of EUS-FNA NTS cases of PDAC and analyzed their management and outcome. Up to September 2022, the search query retrieved forty-five cases plus an unpublished case from our center, for a total of forty-six; 43.6% were male, with a mean age of 68.6 years. Thirty-four patients (87.1%) underwent an initial surgical resection, with only 44.1% and 5.9% undergoing adjuvant and neoadjuvant chemotherapy, respectively, and 5.9% undergoing both. The NTS nodule was mostly located in the posterior gastric wall, developing at a median of 19 months after primary resection; 82.4% underwent surgical resection of the seeding, while for 17.6%, palliative chemotherapy treatment. Follow-up after NTS diagnosis and treatment was reported for only twenty-three patients: when NTS was treated with surgery, the median overall survival was 26.5 months compared to 15.5 if treated with radio/chemotherapy. NTS after EUS-FNA of PDAC occurs late and might be treated aggressively with good results. Interestingly, only a low number of patients developing NTS underwent chemotherapy for the primary cancer, suggesting its possible protective role.

## 1. Introduction

Pancreatic ductal adenocarcinoma (PDAC) is the fourth leading cause of cancer-related mortality [1], and it is estimated to become the second by 2030 [2]. One of the causes of PDAC’s aggressiveness and high recurrence rate is the ability of tumor cells to adapt to a hostile milieu with reduced oxygen and nutrient availability, thus, spreading early in a “micro metastatic” fashion [3]. 

It is, therefore, not surprising that the seeding of PDAC cells is possible. This phenomenon was first reported after percutaneous radiologically-guided fine-needle aspiration (FNA) [4], in the catheter tract of transhepatic percutaneous biliary drainage [5], and, later on, along the path of surgical drainages [6] or trocar insertion.

Endoscopic ultrasound (EUS)-guided FNA is a well-established method for the tissue acquisition of pancreatic neoplasms, with a sensitivity of 85% and a specificity of 98% [7]. Peri-procedural complications include acute pancreatitis, bleeding, perforation, and post-procedural pain, which are usually mild to moderate and develop in less than 1% of cases [8,9]. With such a high diagnostic yield and low risk of complications, EUS-FNA is currently the gold standard for the cytological and histological diagnosis of solid pancreatic lesions. After the development of EUS with pancreatic tissue acquisition in the late 90s [10], a few authors described cases of seeding as long-term complication of EUS-FNA, with the first reports dated 2005 [11], but the risk of needle-tract seeding (NTS) compared to the percutaneous route was lower [12]. Indeed, the European Society of Medical Oncology (ESMO) guidelines for PDAC diagnosis strictly recommend tissue acquisition through EUS-FNA for non-metastatic lesions [13] in order to reduce the risk of peritoneal seeding of percutaneous biopsies. Despite a low estimated prevalence of NTS between 0.003%–0.009% [14] and 0.3% [15], it has been debated whether EUS-FNA should or should not be performed on patients with potentially resectable PDAC [16]. Several studies, however, showed that its use does not affect overall and recurrence-free survival [17,18,19]. A recent meta-analysis on this topic by Facciorusso et al. [15] aimed at investigating both the incidence of NTS and whether EUS-FNA is associated with an increased NTS risk. In this regard, their aim and methodology differ from the present study. Notably, they reported an NTS rate of 0.4% in patients with PDAC and did not find EUS-FNA to be associated with peritoneal dissemination with a rate similar to that of non-sampled patients [15].

Nonetheless, the NTS of PDAC in general is a poorly investigated entity, and the management of the seeding localization has been scarcely discussed. While the first reports would consider NTS similarly to disease recurrence or metastatic progression of the disease, offering only palliative support to the patient, the most recent ones started considering a surgical eradication of the new disease site [17]. This extreme variety in disease management underlines the lack of knowledge on this entity and its natural history.

Also, given the increase in both the incidence of PDAC and the use of EUS-FNA, NTS recognition and the development of its standardized management seem necessary. 

Another relevant aspect is the association of NTS with perioperative treatments. As the rationale for both adjuvant and neoadjuvant chemotherapy of PDAC is to reduce the risk of micro-metastases presenting as distant recurrence sites after surgery, it is possible to hypothesize that they also reduce the risk of NTS. This is, however, an unexplored issue.

The aim of our study was, therefore, to conduct a systematic review of cases of NTS of PDAC associated with the use of EUS-FNA and analyze their clinical features, management, and outcome, including those of an unpublished report from our center.

## 2. Materials and Methods

### 2.1. Systematic Literature Search

Adhering to 2020 PRISMA guidelines [20], a systematic search was performed in PubMed and Scopus, for case reports and series concerning needle tract seeding of pancreatic cancer published up to 19 September 2022. The following search terms were used: (pancreas OR pancreatic OR pancreas cancer OR pancreatic cancer OR pancreas tumor OR pancreatic tumor) AND (seeding).

Two authors (LA and RPDLP) screened all titles and abstracts for eligible articles based on predefined eligibility criteria. Full-text manuscripts were evaluated, and references of the identified manuscripts were examined for further potentially relevant articles not identified by the initial search.

This review has been registered in PROSPERO–National Institute for Health Research, ID –CRD42022356314.

### 2.2. Eligibility Criteria

Inclusion criteria were: (a) case reports or case series concerning patients with a pancreatic lesion undergoing EUS with FNA/FNB; (b) development of a lesion that is highly suspect or cyto/histologically confirmed as deriving from the primary pancreatic tumor; (c) the lesion is described as being located in the trajectory of the previous FNA puncture site, and therefore, highly suggestive for NTS. Exclusion criteria were: (a) pancreatic lesions different from PDAC; and (b) peritoneal recurrences.

In the event of duplicate publications, the most recent or more complete one was selected.

### 2.3. Data Extraction

The following data were retrieved and registered in a dedicated spreadsheet: country of origin, publication year, patient gender and age, PDAC-EUS and surgical specimen size, site, caliber and type of needle used to perform FNA on the primary PDAC, number of FNA passes, puncture route, TNM stage and management of the primary PDAC, histological diagnosis, grading and R status of the resection, localization of seeding, time lapse between EUS-FNA of the primary tumor and diagnosis of seeding nodule, seeding nodule management, and patient outcome. Whenever any of this information was not provided in the manuscript, we attempted to retrieve them by contacting the corresponding author by email.

A case of NTS diagnosed at our center was also included.

## 3. Results

### 3.1. Case Report

We report the case of a 57-year-old male, referred to our center for the follow-up of alcoholic chronic pancreatitis, who underwent pancreatico-jejunostomy in 2015 for recurrent acute flares. In March 2016, at a follow-up EUS, a 20 mm mass of the pancreatic tail was diagnosed. An EUS-FNA of the lesion with a 25-gauge Menghini-tip needle (Expect SlimLine, Boston Scientific, MA, USA) was carried out through the posterior gastric wall (2 passes) with a slow-pull technique, and the cytological examination revealed PDAC. The patient had a Karnofsky performance status score of up to 80%, and the pre-operative computed tomography (CT) scan confirmed the 20 × 20 mm lesion of the tail, invading the splenic vessels; levels of Ca19.9 were slightly elevated (175 U/ml). In April 2016, the patient underwent upfront distal pancreatectomy with splenectomy for T2N0M0 R0 G3 (TNM 8th edition) PDAC, with 14 negative lymph nodes removed. Follow-up management included adjuvant chemotherapy with 6 cycles of gemcitabine and radiological surveillance with CT scan every 3 months.

A CT scan performed 9 months after surgical resection showed the onset of a focal mass in the posterior gastric wall. EUS revealed a 25 mm hypoechogenic mass arising from the muscularis propria layer of the posterior gastric wall, stiff at elastography, isoenhanced after the intravenous injection of SonoVue^®^ (Bracco, Milan, Italy) contrast agent with a hypoenhanced core (Figure 1C). Initial differential diagnosis was between a malignant, although small, gastrointestinal stromal tumor (GIST) and a case of seeding due to the previous EUS-FNA. The cytological examination of the EUS-FNA tissue acquired reported a localization of PDAC.

The patient underwent atypical gastric resection in April 2017 with a histological confirmation of a 35 mm gastric recurrence of the previous PDAC from the submucosal layer to beyond the serosa. The mass was located in the site of the previous EUS-FNA puncture, strongly suggesting a EUS-FNA NTS.

He subsequently underwent further adjuvant chemotherapy, with evidence of liver metastases onset at 6-month CT scan, and patient exitus on April 2018, 25 months after the first diagnosis of PDAC.

### 3.2. Results of the Systematic Review

#### 3.2.1. Selection of Included Studies

The search initially retrieved 1374 results from Pubmed and 326 results from Scopus. After a screening of titles and abstracts, 30 reports (either case reports or series) were selected from Pubmed with no additional reports from Scopus. Another 3 reports were identified from hand-searching of the references of the selected manuscripts and included, for a total of 33 reports [11,17,19,21,22,23,24,25,26,27,28,29,30,31,32,33,34,35,36,37,38,39,40,41,42,43,44,45,46,47]. (Figure 2).

As there were some missing data in all the retrieved studies, we tried to retrieve them by contacting all corresponding authors of these 33 papers, but we only managed to receive a reply from three of them [19,44,45], although the requested data were available for only one of them. We did not include a recent Japanese nationwide survey [48] investigating the status of NTS after the EUS-FNA of primary pancreatic tumors in surgically resected patients because it did not satisfy the eligibility criteria; however, we analyzed its results in the discussion.

#### 3.2.2. Features of the Included NTS Cases

The unpublished newly developed case from our center was added to the 45 literature cases from the 33 reports for a total of 46 cases, described between 2005 and 2022 [11,17,19,21,22,23,24,25,26,27,28,29,30,31,32,33,34,35,36,37,38,39,40,41,42,43,44,45,46,47].

Seventeen of the 46 patients (43.6%) were male, with a mean age of 68.6 years. Forty-three cases (93.5%) arose from the pancreatic body/tail, 3 (6.5%) from the pancreatic head; all 46 cases (100%) were histologically diagnosed as Pancreatic Ductal Adenocarcinoma (PDAC).

The caliber of the needles used to acquire tissue for the primary cancer diagnosis was 22 G in 27 cases (64.3% of cases), but other needles of 19, 20, 21, or 25 G were also employed; in terms of number of needle passes, these were >1 in 33 (94.3%) of the reported cases and varied between 1 and 5. Puncture route was trans-gastric in 44 cases (95.6%).

Thirty-four patients (87.1%) underwent surgical resection of the primary pancreatic lesion, of which only 2 (5.9%) received neoadjuvant chemotherapy, 15 (44.1%) received adjuvant chemotherapy, and 2 (5.9 %) received both. Five patients (12.8%) were not resected and were treated with primary chemotherapy and/or radiation therapy. Thus, the majority of the reported NTS cases were treated with surgery alone on the primary tumor.

Time between EUS-FNA and the diagnosis of NTS ranged between a few days to a maximum of 42 months, with a median onset after 19 months. In 42 (91.3%) patients, NTS was reported in the gastric wall, whereas 2 (4.4%) were reported in the remaining pancreas itself, 1 (2.15%) in the duodenum, and 1 (2.15%) in the cardia.

#### 3.2.3. Treatment of the Seeding Site and Outcomes

Concerning the management, 28 (82.4%) patients diagnosed with NTS underwent a new surgical resection as treatment for the seeding nodule, mostly localized in the posterior gastric wall, while 6 (17.6%) underwent palliative chemotherapeutic treatment, being, therefore, managed as a metastatic spread of the disease instead. For 12 patients (26.1%), no information about NTS treatment was available, and for 24 patients (52.1%), no information about overall survival was available. The corresponding authors of 20 of these patients were contacted, but replies for only 10 patients were received, with most of them replying that no information could be retrieved except for one case [44].

Of the 28 patients undergoing surgical resection of the gastric NTS, disease re-recurrence after NTS removal was described in 6 patients (1 in the liver, 2 in both liver and peritoneal, 2 peritoneal, and 1 had a second gastric recurrence). 

The median overall survival of NTS treated with surgery was 26.5 months compared to the 15.5 months of survival of patients treated with oncological treatment alone. Although a statistical comparison was not considered methodologically sound, an aggressive treatment seemed to be associated with better outcome.

Other information about chemotherapy, surgical treatment, EUS-FNA, seeding nodule, tumor stage, and patient management are reported in Table 1.

## 4. Conclusions

NTS is a rare but serious long-term complication of EUS-FNA, with a very low prevalence that has been reported to range between 0.003% and 0.009% [14]. A recent systematic review and meta-analysis by Facciorusso et al. [15] calculated this incidence to be higher, up to 0.4% when focusing on pancreatic adenocarcinoma, albeit without an apparent impact on prognosis, as shown by the similar rate of carcinomatosis onset reported in patients undergoing EUS-FNA or not. In that meta-analysis, nonetheless, the management of NTS and how this could impact the prognosis was investigated. 

With the widespread use of EUS with tissue acquisition for pancreatic lesions, NTS localizations may be expected to become more frequent, and their standardized management is therefore advisable. The management of NTS localization is, instead, not standardized and several options, from surgical resection to palliative chemotherapy could be considered. 

The present systematic review reported data on 46 cases that fulfilled our stringent inclusion criteria for cases in which PDAC seeding on the needle track was deemed extremely likely. In some of these cases, the authors performed molecular analyses of the NTS tissue and compared it to the primary pancreatic tumor to confirm the etiology of the nodule based on the striking similarity of genomic landscape [37]. We excluded cases presenting with seeding in the peritoneum. In case of carcinomatosis onset, it could indeed be difficult to distinguish between procedure-related seeding and disease metastatic dissemination. In keeping with our approach, the study by Facciorusso et al. proved that the use of EUS-FNA does not seem to increase the rate of peritoneal carcinomatosis [15], and therefore, it is unlikely that cases of NTS with peritoneal carcinomatosis are genuine seeding cases.

We gathered information to homogenize available data from previous sparse reports and hereby present the largest combined analysis on NTS features in 46 patients. Although our aim was to retrieve all of the cases’ missing data by contacting the authors, the long time gap between the first and the last reported cases did not allow us to obtain complete information about the clinical course.

Most of the 46 included NTS cases were located in the gastric wall (91.3%), a minority being in the duodenum, in the cardia, and in the pancreas itself. This is not surprising, as most of the seeding cases occurred after trans-gastric biopsies of lesions of the body-tail that would have their stomach preserved when surgically resected [11,17,19,21,22,23,25,26,27,28,29,30,31,32,33,34,35,36,37,38,39,40,41,42,45,46]. On the other hand, in cases of lesions in the pancreatic head, the eventual onset of seeding could remain undetected, as the wall pierced by FNA route (duodenal bulb or second portion) would be removed during surgery [43], but no microscopic pathological investigation would be performed on such site; second, considering the short overall survival of PDAC patients [1], NTS foci may not have enough time to emerge and become detectable during follow-up investigations. 

One may therefore hypothesize that the real incidence of NTS in general may be underestimated and underdiagnosed.

An interesting issue arose regarding the timing of NTS diagnosis: there was a wide range, from few days to 42 months (median 19 months), even in patients with similar disease course and treatments. This heterogeneity is likely explained by the variability in terms of growth rate that is also observed in primary PDAC treated with chemotherapy or experiencing recurrence after radical surgery [13]. In this view, NTS might be considered an interesting model of PDAC cell-growth patterns and timeline and help clarify the pathophysiology of disease recurrence. Indeed, how much time it takes for PDAC to grow from single cells to a solid nodule until symptom onset remains largely uncertain.

In terms of NTS management, it is interesting to highlight how in a minority of the reported cases (17.6%), NTS was considered and, therefore, managed as a metastatic site of disease, usually bearing an expected survival of <12 months and, therefore, receiving either chemotherapy or palliative treatments alone. Most patients (82.4%) were instead treated with secondary surgical excision of the NTS nodule, as their NTS was considered as a de novo T2N0M0 lesion [17]. This more aggressive approach succeeded in eradicating the recurrent disease in some cases. The lack of data about the patients’ overall survival and the small number of reported cases did not, however, allow meaningful comparison between different treatments and their outcomes, despite NTS treated with surgery showing a longer median overall survival (26.5 months) compared to those treated with chemotherapy or palliation (15.5 months). 

A very recent nationwide survey conducted in Japan [48] reported data about NTS in 12,109 patients who underwent surgical resection of primary pancreatic tumors. In that study, the NTS incidence (40 patients–0.33%) was low and localization (97.4%) was mostly in the gastric wall. Twenty-five patients (65.8%) underwent surgical resection of the seeding lesion, with a longer overall survival compared to those who did not. Most data from that survey seem, therefore, in keeping with the present study. Unfortunately, the lack of information about neoadjuvant or adjuvant chemotherapy in that cohort strongly affects the clinical relevance of the results. Also, as the data was only derived from a survey, with a different methodology and lack of detailed information, we did not include these cases in our qualitative analysis of the literature.

The reported cases of NTS support the concern of EUS being potentially risky for the diagnosis and tissue acquisition of resectable pancreatic neoplasms. Some researchers have indeed highlighted a possible role of EUS-FNA in promoting distant metastasis, analyzing cell-free DNA plasma concentration before and after EUS-FNA of PDAC and showing how EUS-FNA is associated with increased plasma concentration of cell-free DNA and increased detection of mutant KRAS [49]. As a matter of fact, many NTS cases were located in the gastric submucosa, a layer characterized by abundant blood and lymphatic vessels, supporting the pathogenic mechanism of NTS in promoting metastatic spread [44]. Nonetheless, several studies have demonstrated that EUS-FNA of PDAC is associated with neither worse overall nor recurrence-free survival [17,18,19]. 

Since the first publication of reports about NTS after EUS-FNA, several novel methods and devices have been introduced to reduce the number of required punctures [13]. In the present cohort, 22 G was the most commonly employed needle (64.3%), and almost all patients underwent more than 1 pass (94.3%). Interestingly, it has been advocated to minimize the number of FNA/FNB passes to reduce the risk of complications, but possibly also to reduce the risk of seeding, for example, with the use of rapid onsite evaluation(ROSE) when available, or with the use of FNB needles to acquire more tissue with less passes. Furthermore, an additional suggestion could be to wipe the needle, flushing it with sterile saline, between each pass to remove any remaining tumoral cell. Although there are some reports of confirmed NTS after single passes [17], the majority of published cases reported NTS to occur after multiple passes; considering the low number of studies on the topic, nonetheless, the association between a defined needle or the number of passes and NTS onset as a complication cannot be ascertained. Notably, an additional interesting aspect regarding the FNA technique applied was observed: the “slow-pull” technique applies, in fact, less pressure compared with the “suction” one, possibly leading to an easier cell fall in the needle route and, therefore, a higher risk of seeding. However, as only few of the reports described the technique of the biopsy, a comparative analysis was not possible. 

While the event of seeding during EUS for PDAC should be kept in mind, we might speculate that this risk may be reduced by neoadjuvant and adjuvant treatments. In fact, notably, only 55.9% of the patients with reported NTS who underwent surgical resection of the primary pancreatic tumor had undergone neoadjuvant and/or adjuvant chemotherapy for the primary tumor, despite current guidelines recommending a perioperative chemotherapy for all fit patients [50]. One may speculate that isolated tumor cells may be easily eliminated by an active chemotherapy administered either before or soon after surgery, reducing the potential risk of NTS. The wider diffusion of neoadjuvant polychemotherapy before surgery [2] will, therefore, likely reduce NTS incidence. Large studies comparing the rate of NTS in patients treated or not with different perioperative chemotherapy policies (none, adjuvant, neoadjuvant, and both) are needed to confirm this hypothesis.

In conclusion, the present systematic review gathered available data on NTS after EUS-FNA pooling the largest dataset of reported cases to date, with data confirming that this rare complication usually occurs late (median onset after 19 months from EUS-FNA) and should be distinguished from “typical” distant disease recurrence since, when treated more aggressively with repeated surgery, the overall survival seems longer (26.5 months) compared to that of patients treated with palliative treatments (15.5 months).

## Figures and Tables

**Figure 1 cancers-14-06130-f001:**
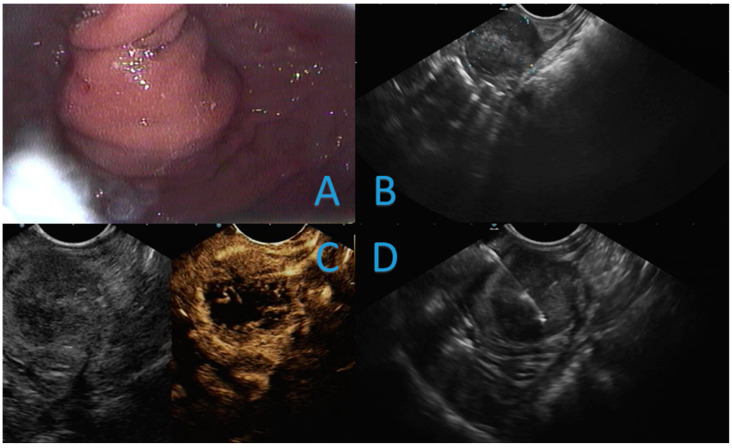
Endoscopic and EUS images of the case from our Center. Panel (**A**) endoscopic view of the gastric bulging of the seeding; Panel (**B**) EUS measurement of the seeding nodule; Panel (**C**) contrast enhancement of the seeding nodule; Panel (**D**) fine-needle aspiration of the lesion.

**Figure 2 cancers-14-06130-f002:**
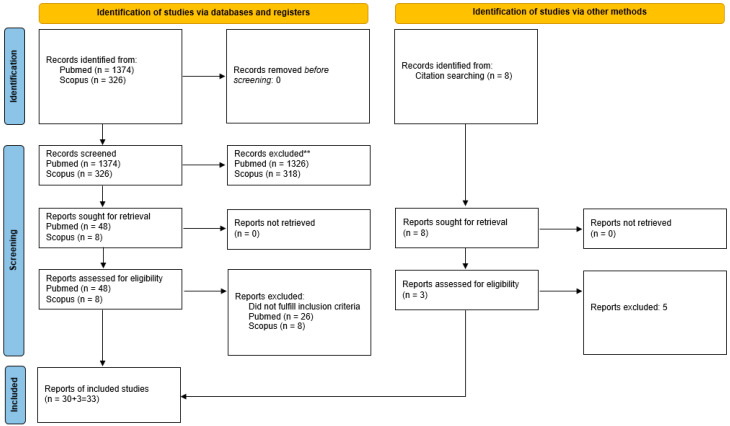
PRISMA 2020 flow diagram of the systematic search.

**Table 1 cancers-14-06130-t001:** Clinical features of the 36 cases of needle-tract seeding (NTS).

Number	Author	Year	Diagnosis	Age at PDAC Diagnosis	Tumor Site	Stage	Needle (Gauge)	Puncture Route	Needle Passes	Interval from EUS-FNA (Months)	Seeding Site	Primary Tumor Management	NTS Management	Overall Survival (Months)
1	Paquin et al. [11]	2005	PDAC	65	PT	T1N0M0	22	TG	5	21	PGW	S	CHT (PALL)	33
2	Ahmed et al. [21]	2011	PDAC	79	PB	T2N0M0	NA	TG	Several	39	GW	S + ADJ CHT-RT	S (G)	36
3	Chong et al. [22]	2011	PDAC	55	PT	T2N0M0 R0	22	TG	2	26	PGW	S	PALL	NA
4	Katanuma et al. [23]	2012	PDAC	68	PB	T2N0M0	22	TG	4	22	PGW	S	NA	NA
5	Anderson et al. [24]	2013	PDAC	51	PH, CL	NA	NA	GEJ	Several	NA	GEJ	CHT-RT	NA	NA
6	Ngamruengphong et al. [19]	2013	PDAC	66	PB, PT	IIA	19, 22	TG	3	27	GW	S + ADJ CHT-RT	NA	NA
7	Ngamruengphong et al. [19]	2013	PDAC	77	PT	IIA	19	TG	3	26	GW	S + ADJ CHT-RT	NA	NA
8	Minaga et al. [25]	2015	PDAC	64	PB	T3N0M0	22	TG	3	8	PGW	S	S (PG)	NA
9	Naruse H et al. [26]	2015	PDAC	72	PB	NA	NA	TG	4	9	GW	CHT	CHT (PALL)	11
10	Tomonari et al. [27]	2015	PDAC	78	PB	T3N0M0 R0	22	TG	2	9	PGW	S + ADJ CHT	S (PG)	NA
11	Kita et al. [28]	2016	PDAC	68	PB, PT	NA	22	TG	2	4	PGW	RT	NA	NA
12	Minaga et al. [29]	2016	PDAC	72	PB	T1N0M0 R0	NA	TG	NA	24	PGW	S	S (G)	NA
13	Yamauchi et al. [30]	2016	PDAC	65	PB	T3N0M0	22	TG	1	22	PGW	S	S (PG)	NA
14	Iida et al. [31]	2016	PDAC	65	PT	T3N0M0	22	TG	3	6	PGW	S	S (PG) + CHT (ADJ)	27
15	Yamabe et al. [32]	2016	PDAC	75	PB	NA	25	TG	NA	3	PGW	CHT (PALL)	CHT (PALL)	29
16	Yasumoto et al. [33]	2018	PDAC	78	PB	T3N0M0 R0	25	TG	NA	22	PGW	S + ADJ CHT	S (PG)	NA
17	Matsumoto et al. [34]	2018	PDAC	50	PB	T3N0M0	21	TG	3	8	PGW	NEOADJ CHT + S	Included in the tumor management	NA
18	Sakamoto et al. [35]	2018	PDAC	50	PT	T4N1M0	22	TG	2	24	PGW	S + ADJ CHT	S (PG)	NA
19	Matsui et al. [36]	2019	PDAC	68	PB	T1N1M0	19, 20, 22	TG	4	1	PGW	S + ADJ CHT	Included in the tumor management	18
20	Matsui et al. [36]	2019	PDAC	70	PB	T2N0M1	22	TG	1	4	PGW	NEOADJ CHT + S + ADJ CHT	Included in the tumor management	18
21	Kawabata et al. [37]	2019	PDAC	78	PB	I R0	22	TG, TP	NA	36	PGW	S	S (PG)	NA
22	Yane et al. [38]	2020	PDAC	66	PT	T3N0M0 G1 R0	22	TG	4	18	PGW	S	CHT (PALL)	10
23	Yane et al. [38]	2020	PDAC	78	PT	T3N0M0 G3 R1	22	TG	2	26	PGW	S	S	25
24	Yane et al. [38]	2020	PDAC	86	PB	T2N0M0 G3 R0	22	TG	3	18	PGW	S	S	62
25	Yane et al. [38]	2020	PDAC	47	PB	T2N0M0 G2 R0	22	TG	4	27	PGW	S	S	17
26	Yane et al. [38]	2020	PDAC	79	PB	T1N0M0 G3 R0	22	TG	3	6	PGW	S	S	40
27	Yane et al. [38]	2020	PDAC	78	PB	T1N0M0 G2 R0	22	TG	4	4	PGW	S	S	5
28	Sato et al. [39]	2020	PDAC	83	PB	T2N2M0 R0	22	TG	2	25	PGW	S + ADJ CHT	S (PG)	30
29	Rothermel et al. [40]	2020	PDAC	61	PB	T3N0M0 R0	25	TG	3	42	PGW	S + ADJ CHT	NEOADJ CHT + S (PG) + ADJ CHT	72
30	Hayasaka et al. [41]	2020	PDAC	75	PT	T1N0M0	22	TG	3	2	GW	S + ADJ CHT	S (PG)	NA
31	Okamoto et al. [42]	2020	PDAC	72	PT	T3N1M0 R0	22	TG	5	0.3	PGW	NEOADJ CHT + S + ADJ CHT	Included in the tumor management	9
32	Kojima et al. [17]	2021	PDAC	81	PT	T1N1M1 R0	22	TG	4	0.1	PGW	S + ADJ CHT	Included in the tumor management	8
33	Nakatsubo et al. [43]	2021	PDAC	67	PH	IIB R1	22	TD	2	0	D	S	Included in the tumor management	NA
34	Uozumi et al. [44]	2021	PDAC	77	PT	T2N0M0	22	TG	2	0	Pancreas	S + ADJ CHT	Included in the tumor management	26
35–41	Kanno et al. [45]	2021	PDAC	NA	PT	NA	21, 22, 25	TG	NA	NA	PGW	NA	NA	NA
42	Lovecek et al. [46]	2022	PDAC	65	PT	T3N1M0 R0 G1	22	TG	2–4	33	PGW	NEOADJ + RT + S	S (PG) + ADJ CHT	56
43	Lovecek et al. [46]	2022	PDAC	71	PT	T1N0M0 R0 G2	22	TG	2–4	27	PGW	S + ADJ CHT	S (PG) + ADJ CHT	82
44	Lovecek et al. [46]	2022	PDAC	75	PH	T3N0M0 R0 G3	22	TG	2–4	19	GW	S + ADJ CHT	S (PG)	28
45	Mizumoto et al. [47]	2022	PDAC	45	PT	NA	22	TG	3	1	Pancreas	CHT (PALL)	Included in the tumor management	20
46	Archibugi et al.	2022	PDAC	57	PT	T2N0M0 R0 G3	25	TG	2	9	PGW	S +ADJ CHT	S (PG) + ADJ CHT	25

Abbreviations: PDAC = pancreatic ductal adenocarcinoma; PT = pancreas tail; PB = pancreas body; PH = pancreas head; TG = trans-gastric; TP = trans-papillary TD = trans-duodenal; D = duodenum; S = surgery; NA = not available; GEJ = gastro-esophageal junction; GW = gastric wall; PGW = posterior gastric wall; G = gastrectomy; PG = partial gastrectomy; CL = coeliac lymph node; ADJ = adjuvant; NEOADJ = neoadjuvant; PALL = palliative; CHT = chemotherapy; RT = radiotherapy.

## Data Availability

The data presented in this study are available in Table 1.

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
