# Peer review of "Needle-Tract Seeding of Pancreatic Cancer after EUS-FNA: A Systematic Review of Case Reports and Discussion of Management"

_cancers, 2022, doi:10.3390/cancers14246130_

Round 1

Reviewer 1 Report (Previous Reviewer 2)

This paper was corrected appropriately, and is worth reading for clinician in gastroenterology.

Author Response

We thank the reviewer for approving the current version of the manuscript and for finding it useful to gastroenterologists.

Reviewer 2 Report (Previous Reviewer 1)

.

Author Response

We thank the reviewer for considering the new version of the paper and giving us some additional suggestions. We have carefully revised the paper accordingly to your suggestions.

This manuscript is a resubmission of an earlier submission. The following is a list of the peer review reports and author responses from that submission.

Round 1

Reviewer 1 Report

Many thanks for asking me to review this paper. It does cover an important topic, however, there has been a recent systematic review and metanalysis of the same topic ( Facciorusso et al Needle Tract Seeding after Endoscopic Ultrasound Tissue Acquisition of Pancreatic Lesions: A Systematic Review and Meta-Analysis. Diagnostics 2022, 12, 2113. https://doi.org/10.3390/ diagnostics12092113) which  authors referenced prevalence of Needle tract seeding (NTS) with no further critical discussion. Therefore, it was very important to review the aim of this study and giving their mentioned aims, unfortunately, it doesn’t add much to the knoweldge compared to what already published. Evidently, more cases did not provide any further deep knowledge. 

Study design: There are major flaws in the design and methodology. Line 249- authors report 17.9% to be reported as metastatic disease but they believe it is NTS. Were these cases reported as NTS or metastatic disease? This would contradict the eligibility criteria with the lack of further information form original reports’ authors, one would question whether they should be included, therefore, further explanation is required.  The first case in table one is a case of peritoneal dissemination and the nodule resected from the stomach did not show malignant cell but the conclusion from author is still NTS in the posterior gastric wall. Line 104- What measures were taken to ascertain recurrences at puncture site?

Suggestion

It looks it is a systematic review of case reports and should be reflected in the title 

There is a lot or repetition for example, but not exhaustive,  the statement on contacting authors. 

Due to low number of cases suggest authors state n (%) throughout.

Line 17- authors make a general statement about a low incidence of NTS but later in discussion it appears it is unknow due to many factors including those at head of pancreas may go undetected which is a common site. 

Line 25- is it a minority? it is mentioned that 48.5% went to have some sort of chemotherapy

Line 40- the statement is unclear- how the conclusion was generated?

Reference 1 – need up to date reference 

Line 48- within 

Line 49- (due to) is redundant

Systematic literature search. Since it is about finding cases report? Would search on another data search engine would’ve given any more?

Result- two important information missing performance status and age.

Line 235. Why the longer path the higher chance of NTS, any technical explanation or any other explanation for the reader

Line 239 The conclusion should be added to the previous paragraph, not standalone

Line 274- authors state that EUS is a risk procedure and can cause cancer metastasis which contradict their statement in line 218

Line 279- this paragraph needs more references as it is a very important area to discuss.

Line 297- does the finding support the use of chemotherapy?

Reviewer 2 Report

This is a well documented manuscript about NTS and worth reading for gastroenterologist and surgens.

However, the conclusion is expected to describe the document associated with the results from the analysis. Moreover, description of the incidence ratio is also expected to be included in the results.

Reviewer 3 Report

This study was aimed to review systematically the fact of needle tract seeding of pancreatic cancer after EUS-FNA. The occurrence of this unusual complications of EUS-FNA in patients with pancreatic cancer is not so common. Therefore, systematic search might be a reasonable approach to reveal the clinical implication of this event. There were some concerns in this manuscript to be described in more details for more clarification and usefulness.

1, The clinical diagnosis of needle tract seeding after EUS-FNA for pancreatic cancer is thought to be difficult sometimes. If the recurrent lesion is localized on the gastric wall where EUS-FNA was previously punctured, the diagnosis of needle tract seeding could be recognized without difficulty.

However, if recurrent lesions locate freely in the abdominal cavity, differential diagnosis between EUS-FNA seeding and peritoneal disseminated recurrence after surgical resection might be very difficult. How did authors judge whether each reported cases were correctly diagnosed as EUS-FNA seeding recurrence? Authors should clarify how to differentiate recurrent lesions between EUS-FNA seeding and just postoperative peritoneal metastatic recurrence.

2, It was described that “needle tract seeding nodules were mostly located in the posterior gastric wall” in this manuscript. Authors should show other recurrent sites other than posterior gastric wall in this systemic review analysis.  

3, Authors analyzed the survival following aggressive re-do surgery for the recurrence of EUS-FNA seeding recurrence, and showed favorable long-term outcome after redo surgery as compared with other non-surgical therapies. It might be important to know which were re-recurrent site after redo surgery for EUS-seeding recurrence. Authors should describe the re-recurrent site after redo surgery in long term follow up if systemic is possible.

4, It was described that the occurrence of EUS-FNA seeding was diagnosed at a median of 19 months after primary diagnosis in this manuscript. However, in this study series, there were some patients who got neoadjuvant therapy for long periods. Therefore, authors should measure intervals of reported cases between EUS-FNA test and the diagnosis of occurrence of EUS-FNA seeding.